# Countering Feedback Delays in Multi-Agent Learning

**Zhengyuan Zhou**
Stanford University
zyzhou@stanford.edu

**Panayotis Mertikopoulos**
Univ. Grenoble Alpes, CNRS, Inria, LIG
panayotis.mertikopoulos@imag.fr

**Nicholas Bambos**
Stanford University
bambos@stanford.edu

**Peter Glynn**
Stanford University
glynn@stanford.edu

**Claire Tomlin**
UC Berkeley
tomlin@eecs.berkeley.edu

## Abstract

We consider a model of game-theoretic learning based on online mirror descent (OMD) with asynchronous and delayed feedback information. Instead of focusing on specific games, we consider a broad class of continuous games defined by the general equilibrium stability notion, which we call $\lambda$-*variational stability*. Our first contribution is that, in this class of games, the actual sequence of play induced by OMD-based learning converges to Nash equilibria provided that the feedback delays faced by the players are synchronous and bounded. Subsequently, to tackle fully decentralized, asynchronous environments with (possibly) unbounded delays between actions and feedback, we propose a variant of OMD which we call delayed mirror descent (DMD), and which relies on the repeated leveraging of past information. With this modification, the algorithm converges to Nash equilibria with no feedback synchronicity assumptions and even when the delays grow superlinearly relative to the horizon of play.

## 1 Introduction

Online learning is a broad and powerful theoretical framework enjoying widespread applications and great success in machine learning, data science, operations research, and many other fields [3, 7, 22]. The prototypical online learning problem may be described as follows: At each round $t = 0, 1, \ldots$, a player selects an action $x^t$ from some convex, compact set, and obtains a reward $u^t(x^t)$ based on some *a priori unknown* payoff function $u^t$. Subsequently, the player receives some feedback (e.g. the past history of the reward functions) and selects a new action $x^{t+1}$ with the goal of maximizing the obtained reward. Aggregating over the rounds of the process, this is usually quantified by asking that the player's (external) *regret* $\text{Reg}(T) \equiv \max_{x \in \mathcal{X}} \sum_{t=1}^{T} [u^t(x) - u^t(x^t)]$ grow sublinearly with the horizon of play $T$, a property known as "no regret".

One of the most widely used algorithmic schemes for learning in this context is the online mirror descent (OMD) class of algorithms [23]. Tracing its origins to [17] for offline optimization problems, OMD proceeds by taking a gradient step in the dual (gradient) space and projecting it back to the primal (decision) space via a *mirror map* generated by a strongly convex regularizer function (with different regularizers giving rise to different algorithms). In particular, OMD includes as special cases several seminal learning algorithms, such as Zinkevich's online gradient descent (OGD) scheme [29], and the multiplicative/exponential weights (EW) algorithm [1, 13]. Several variants of this class also exist and, perhaps unsurprisingly, they occur with a variety of different names – such as "Follow-the-Regularized-Leader" [9], dual averaging [18, 25], and so on.

When $u^t$ is concave, OMD enjoys a sublinear $\mathcal{O}(\sqrt{T})$ regret bound which is known to be universally tight.[1] A common instantiation of this is found in repeated multi-player games, where each player's payoff function is determined by the actions of all other players via a fixed mechanism – the *stage game*. Even though this mechanism may be unknown to the players, the universality of the OMD regret bounds raises high expectations in terms of performance guarantees, so it is natural to assume that players adopt some variant thereof when faced with such online decision processes. This leads to the following central question: *if all players of a repeated game employ an OMD updating rule, do their actions converge to a Nash equilibrium of the underlying one-shot game?*

**Related Work.** Given the prominence of Nash equilibrium as a solution concept in game theory (compared to coarser notions such as correlated equilibria or the Hannan set), this problem lies at the heart of multi-agent learning [4]. However, convergence to a Nash equilibrium is, in the words of [4], "considerably more difficult" than attaining a no-regret state for all players (which leads to weaker notion of coarse correlated equilibrium in finite games). To study this question, a growing body of literature has focused on special classes of games (e.g. zero-sum games, routing games) and established the convergence of the so-called "ergodic average" $T^{-1}\sum_{t=1}^{T} x^t$ of OMD [2, 10, 12].

In general, the actual sequence of play may fail to converge altogether, even in simple, finite games [16, 24]. On the other hand, there is a number of recent works establishing the convergence of play in potential games with finite action sets under different assumptions for the number of players involved (continuous or finite) and the quality of the available feedback (perfect, semi-bandit/imperfect, or bandit/payoff-based) [5, 11, 14, 19]. However, these works focus on games with *finite* action sets and feedback is assumed to be *instantly* available to the players (i.e. with no delays or asynchronicities), two crucial assumptions that we do not make in this paper.

A further major challenge arises in decentralized environments (such as transportation networks), where a considerable delay often occurs between a player's action and the corresponding received feedback. To study learning in such settings, [20] recently introduced an elegant and flexible delay framework where the gradient at round $t$ is only available at round $t + d^t - 1$, with $d^t$ being the delay associated with the player's action at round $t$.[2] [20] then considered a very natural extension of OMD under delays: updating the set of gradients as they are received (see Algorithm 1 for details). If the total delay after time $T$ is $D(T) = \sum_{t=1}^{T} d^t$, [20] showed that OMD enjoys an $\mathcal{O}(D(T)^{1/2})$ regret bound. This natural extension has several strengths: first, no assumption is made on how the gradients are received (the delayed gradients can be received out-of-order); further, as pointed out in [6, 8], a gradient "does not need to be timestamped by the round $s$ from which it originates," as required for example by the pooling strategies of [6, 8].

**Our Contributions.** Our investigations here differ from existing work in the following aspects: First, we consider learning in games with asynchronous and delayed feedback by extending the general single-agent feedback delay framework introduced in [20]. Previous work on the topic has focused on the regret analysis of single-agent learning with delays, but the convergence properties of such processes in continuous games are completely unknown. Second, we focus throughout on the convergence of the actual sequence of play generated by OMD (its "last iterate" in the parlance of optimization), as opposed to the algorithm's ergodic average $\frac{1}{T}\sum_{t=1}^{T} x^t$. This last point is worth emphasizing for several reasons: *a)* this mode of convergence is stronger and theoretically more appealing because it implies ergodic convergence; *b)* in a game-theoretic setting, payoffs are determined by the actual sequence of play, so ergodic convergence diminishes in value if it is not accompanied by similar conclusions for the players' realized actions; and *c)* because there is no inherent averaging, the techniques used to prove convergence of $x^t$ provide a much finer understanding of the evolution of OMD.

The starting point of our paper is the introduction of an equilibrium stability notion which we call $\lambda$-*variational stability*, a notion that is motivated by the concept of *evolutionary stability* in population games and builds on the characterization of stable Nash equilibria as solutions to a Minty-type variational inequality [15]. This stability notion is intimately related to monotone operators in variational analysis [21] and can be seen as a strict generalization of operator monotonicity in the

current game-theoretic context.[3] By means of this notion, we are able to treat convergence questions in general games with continuous action spaces, without having to focus on a specific class of games – such as concave potential or strictly monotone games (though our analysis also covers such games).

Our first result is that, assuming variational stability, the sequence of play induced by OMD converges to the game's set of Nash equilibria, provided that the delays of all players are synchronous and bounded (see Theorems 4.3 and 4.4). As an inherited benefit, players adopting this learning algorithm can receive gradients out-of-order and do not need to keep track of the timestamps from which the gradients originate. In fact, even in the special case of learning *without* delays, we are not aware of a similar convergence result for the actual sequence of play.

An important limitation of this result is that delays are assumed synchronous and bounded, an assumption which might not hold in large, decentralized environments. To lift this barrier, we introduce a modification of vanilla OMD which we call *delayed mirror descent* (DMD), and which leverages past information repeatedly, even in rounds where players receive no feedback. Thanks to this modification, play under DMD converges to variationally stable sets of Nash equilibria (Theorem 5.2), even if the players experience asynchronous and unbounded delays: in particular, delays could grow superlinearly in the game's horizon, and DMD would still converge.

We mention that the convergence proofs for both OMD and DMD rely on designing a particular Lyapunov function, the so-called $\lambda$-*Fenchel coupling* which serves as a "primal-dual divergence" measure between actions and gradient variables. Thanks to its Lyapunov properties, the $\lambda$-Fenchel coupling provides a potent tool for proving convergence and we exploit it throughout. Further, we present a unified theoretical framework that puts the analysis of both algorithms under different delay assumptions on the same footing.

## 2  Problem Setup

### 2.1  Games with Continuous Action Sets

We start with the definition of a game with continuous action sets, which serves as a stage game and provides a reward function for each player in an online learning process.

**Definition 2.1.** A *continuous game* $\mathcal{G}$ is a tuple $\mathcal{G} = (\mathcal{N}, \mathcal{X} = \prod_{i=1}^{N} \mathcal{X}_i, \{u_i\}_{i=1}^{N})$, where $\mathcal{N}$ is the set of $N$ *players* $\{1, 2, \ldots, N\}$, $\mathcal{X}_i$ is a compact convex subset of some finite-dimensional vector space $\mathbb{R}^{d_i}$ representing the *action space* of player $i$, and $u_i \colon \mathcal{X} \to \mathbb{R}$ is the $i$-th player's *payoff function*.

Regarding the players' payoff functions, we make the following assumptions throughout:

1. For each $i \in \mathcal{N}$, $u_i(\mathbf{x})$ is continuous in $\mathbf{x}$.

2. For each $i \in \mathcal{N}$, $u_i$ is continuously differentiable in $x_i$ and the partial gradient $\nabla_{x_i} u_i(\mathbf{x})$ is Lipschitz continuous in $\mathbf{x}$.

Throughout the paper, $\mathbf{x}_{-i}$ denotes the joint action of all players but player $i$. Consequently, the joint action[4] $\mathbf{x}$ will frequently be written as $(x_i, \mathbf{x}_{-i})$. Two important quantities in the current context are:

**Definition 2.2.** We let $\mathbf{v}(\mathbf{x})$ be the profile of the players' *individual payoff gradients*,[5] i.e. $\mathbf{v}(\mathbf{x}) = (v_1(\mathbf{x}), \ldots, v_N(\mathbf{x}))$, where $v_i(\mathbf{x}) \triangleq \nabla_{x_i} u_i(\mathbf{x})$.

**Definition 2.3.** Given a continuous game $\mathcal{G}$, $\mathbf{x}^* \in \mathcal{X}$ is called a (pure-strategy) Nash equilibrium if for each $i \in \mathcal{N}$, $u_i(x_i^*, \mathbf{x}_{-i}^*) \geq u_i(x_i, \mathbf{x}_{-i}^*), \forall x_i \in \mathcal{X}_i$.

### 2.2  Online Mirror Descent in Games under Delays

In what follows, we consider a general multi-agent delay model extending the single-agent delay model of [20] to the multi-agent learning case. At a high level, for each agent there can be an arbitrary

delay between the stage at which an action is played and the stage at which feedback is received about said action (typically in the form of gradient information). There is no extra assumption imposed on the feedback delays – in particular, feedback can arrive out-of-order and in a completely asynchronous manner across agents. Further, the received feedback is not time-stamped – so the player might not know to which iteration a specific piece of feedback corresponds.

When OMD is applied in this setting, we obtain the following scheme:

---

**Algorithm 1** Online Mirror Descent on Games under Delays

---

1: Each player $i$ chooses an initial $y_i^0$.
2: **for** $t = 0, 1, 2, \ldots$ **do**
3:     **for** $i = 1, \ldots, N$ **do**
4:         $x_i^t = \arg\max_{x_i \in \mathcal{X}_i}\{\langle y_i^t, x_i \rangle - h_i(x_i)\}$
5:         $y_i^{t+1} = y_i^t + \alpha^t \sum_{s \in \mathcal{G}_i^t} v_i(\mathbf{x}^s)$
6:     **end for**
7: **end for**

---

Three comments are in order here. First, each $h_i$ is a regularizer on $\mathcal{X}_i$, as defined below:

**Definition 2.4.** Let $\mathcal{D}$ be a compact and convex subset of $\mathbb{R}^m$. We say that $g \colon \mathcal{D} \to \mathbb{R}$ is a regularizer if $g$ is continuous and strongly convex on $\mathcal{D}$, i.e. there exists some $K > 0$ such that

$$g(t\mathbf{d} + (1-t)\mathbf{d}') \leq tg(\mathbf{d}) + (1-t)g(\mathbf{d}') - \frac{1}{2}Kt(1-t)\|\mathbf{d}' - \mathbf{d}\|^2 \tag{2.1}$$

for all $t \in [0, 1]$, $\mathrm{bd}, \mathrm{bd}' \in \mathcal{D}$.

Second, the gradient step size $\alpha^t$ in Algorithm 1 can be any positive and non-increasing sequence that satisfies the standard summability assumption: $\sum_{t=0}^{\infty} \alpha^t = \infty, \sum_{t=0}^{\infty} (\alpha^t)^2 < \infty$.

Third, regarding the delay model: in Algorithm 1, $\mathcal{G}_i^t$ denotes the set of rounds whose gradients become available for player $i$ at the current round $t$. Denote player $i$'s delay of the gradient at round $s$ to be $d_i^s$ (a positive integer), then this gradient $v_i(\mathbf{x}^s)$ will be available at round $s + d_i^s - 1$, i.e. $s \in \mathcal{G}_i^{s+d_i^s-1}$. In particular, if $d_i^s = 1$ for all $s$, player $i$ doesn't experience any feedback delays. Note here again that each player can receive feedback out of order: this can happen if the gradient at an earlier round has a much larger delay than that of the gradient at a later round.

## 3    $\lambda$-Variational Stability: A Key Criterion

In this section, we define a key stability notion, called $\lambda$-variational stability. This notion allows us to obtain strong convergence results for the induced sequence of play, as opposed to results that only hold in specific classes of games. The supplement provides two detailed special classes of games (convex potential games and asymmetric Cournot oligopolies) that admit variationally stable equilibria. Other examples include monotone games (discussed later in this section), pseudo-monotone games [28], non-atomic routing games [26, 27], symmetric influence network games [11] and many others.

### 3.1   $\lambda$-Variational Stability

**Definition 3.1.** Given a game with continuous actions $(\mathcal{N}, \mathcal{X} = \prod_{i=1}^N \mathcal{X}_i, \{u_i\}_{i=1}^N)$, a set $\mathcal{C} \subset \mathcal{X}$ is called $\lambda$-*variationally stable* for some $\lambda \in \mathbb{R}_{++}^N$ if

$$\sum_{i=1}^N \lambda_i \langle v_i(\mathbf{x}), x_i - x_i^* \rangle \leq 0, \quad \text{for all } \mathbf{x} \in \mathcal{X}, \mathbf{x}^* \in \mathcal{C}. \tag{3.1}$$

with equality if and only if $\mathbf{x} \in \mathcal{C}$.

*Remark* 3.1. If $\mathcal{C}$ is $\lambda$-stable with $\lambda_i = 1$ for all $i$, it is called simply *stable* [15].

We emphasize that in a game setting, $\lambda$-variational stability is more general than an important concept called operator monotonicity in variational analysis. Specifically, $v(\cdot)$ is called a monotone

operator [21] if the following holds (with equality if and only if $\mathbf{x} = \tilde{\mathbf{x}}$):

$$\langle v(\mathbf{x}) - v(\tilde{\mathbf{x}}), \mathbf{x} - \tilde{\mathbf{x}} \rangle \triangleq \sum_{i=1}^{N} \langle v_i(\mathbf{x}) - v_i(\tilde{\mathbf{x}}), x_i - \tilde{x}_i \rangle \leq 0, \forall \mathbf{x}, \tilde{\mathbf{x}} \in \mathcal{X}. \tag{3.2}$$

If $v(\cdot)$ is monotone, the game admits a unique Nash equilibrium $\mathbf{x}^*$ which (per the property of a Nash equilibrium) satisfies $\langle v(\mathbf{x}^*), \mathbf{x} - \mathbf{x}^* \rangle \leq 0$. Consequently, if $v(\cdot)$ is a monotone operator, it follows that $\langle v(\mathbf{x}), \mathbf{x} - \mathbf{x}^* \rangle \leq \langle v(\mathbf{x}^*), \mathbf{x} - \mathbf{x}^* \rangle \leq 0$, where equality is achieved if and only if $\mathbf{x} = \mathbf{x}^*$. This implies that when $v(\mathbf{x})$ is a monotone operator, the singleton set of the unique Nash equilibrium is $\mathbf{1}$-variationally stable, where $\mathbf{1}$ is the all-ones vector. The converse is not true: when $v(\mathbf{x})$ is not a monotone operator, we can still have a unique Nash equilibrium that is $\lambda$-variationally stable, or more generally, have a $\lambda$-variationally stable set $\mathcal{C}$.

### 3.2 Properties of $\lambda$-Variational Stability

**Lemma 3.2.** *If $\mathcal{C}$ is nonempty and $\lambda$-stable, then it is closed, convex and contains all Nash equilibria of the game.*

The following lemma gives us a convenient sufficient condition ensuring that a singleton $\lambda$-variationally stable set $\{\mathbf{x}^*\}$ exists; in this case, we simply say that $\mathbf{x}^*$ is $\lambda$-variationally stable.

**Lemma 3.3.** *Given a game with continuous actions $(\mathcal{N}, \mathcal{X} = \prod_{i=1}^{N} \mathcal{X}_i, \{u_i\}_{i=1}^{N})$, where each $u_i$ is twice continuously differentiable. For each $\mathbf{x} \in \mathcal{X}$, define the $\lambda$-weighted Hessian matrix $H^\lambda(\mathbf{x})$ as follows:*

$$H_{ij}^\lambda(\mathbf{x}) = \frac{1}{2} \lambda_i \nabla_{x_j} v_i(\mathbf{x}) + \frac{1}{2} \lambda_j (\nabla_{x_i} v_j(\mathbf{x}))^T. \tag{3.3}$$

*If $H^\lambda(\mathbf{x})$ is negative-definite for every $\mathbf{x} \in \mathcal{X}$, then the game admits a unique Nash equilibrium $\mathbf{x}^*$ that is $\lambda$-globally variational stable.*

*Remark* 3.2. It is important to note that the Hessian matrix so defined is a block matrix: each $H_{ij}^\lambda(\mathbf{x})$ is a $d_i \times d_j$ matrix. Writing it in terms of the utility function, we have $H_{ij}^\lambda(\mathbf{x}) = \frac{1}{2} \lambda_i \nabla_{x_j} \nabla_{x_i} u_i(\mathbf{x}) + \frac{1}{2} \lambda_j (\nabla_{x_i} \nabla_{x_j} u_j(\mathbf{x}))^T$.

## 4 Convergence under Synchronous and Bounded Delays

In this section, we tackle the convergence of the last iterate of OMD under delays. We start by defining an important divergence measure, $\lambda$-Fenchel coupling, that generalizes Bregman divergence. We then establish its useful properties that play an indispensable role in both this and next sections.

### 4.1 $\lambda$-Fenchel Coupling

**Definition 4.1.** Fix a game with continuous action spaces $(\mathcal{N}, \mathcal{X} = \prod_{i=1}^{N} \mathcal{X}_i, \{u_i\}_{i=1}^{N})$ and for each player $i$, let $h_i : \mathcal{X}_i \to \mathbf{R}$ be a regularizer with respect to the norm $\|\cdot\|_i$ that is $K_i$-strongly convex.

1. The convex conjugate function $h_i^* : \mathbb{R}^{d_i} \to \mathbf{R}$ of $h_i$ is defined as:

$$h_i^*(y_i) = \max_{x_i \in \mathcal{X}_i} \{\langle x_i, y_i \rangle - h_i(x_i)\}.$$

2. The choice function $C_i : \mathbb{R}^{d_i} \to \mathcal{X}_i$ associated with regularizer $h_i$ for player $i$ is defined as:

$$C_i(y_i) = \arg \max_{x_i \in \mathcal{X}_i} \{\langle x_i, y_i \rangle - h_i(x_i)\}.$$

3. For a $\lambda \in \mathbb{R}_{++}^N$, the $\lambda$-Fenchel coupling $F^\lambda : \mathcal{X} \times \mathbb{R}^{\sum_{i=1}^{N} d_i} \to \mathbf{R}$ is defined as:

$$F^\lambda(\mathbf{x}, \mathbf{y}) = \sum_{i=1}^{N} \lambda_i (h_i(x_i) - \langle x_i, y_i \rangle + h_i^*(y_i)).$$

Note that although the domain of $h_i$ is $\mathcal{X}_i \subset \mathbb{R}^{d_i}$, the domain of its conjugate (gradient space) $h_i^*$ is $\mathbb{R}^{d_i}$. The two key properties of $\lambda$-Fenchel coupling that will be important in establishing the convergence of OMD are given next.

**Lemma 4.2.** *For each $i \in \{1, \dots, N\}$, let $h_i : \mathcal{X}_i \to \mathbb{R}$ be a regularizer with respect to the norm $\|\cdot\|_i$ that is $K_i$-strongly convex and let $\lambda \in \mathbb{R}_{++}^N$. Then $\forall \mathbf{x} \in \mathcal{X}, \forall \tilde{\mathbf{y}}, \mathbf{y} \in \mathbb{R}^{\sum_{i=1}^N d_i}$:*

1. $F^\lambda(\mathbf{x}, \mathbf{y}) \geq \frac{1}{2} \sum_{i=1}^N K_i \lambda_i \|C_i(y_i) - x_i\|_i^2 \geq \frac{1}{2}(\min_i K_i \lambda_i) \sum_{i=1}^N \|C_i(y_i) - x_i\|_i^2$.

2. $F^\lambda(\mathbf{x}, \tilde{\mathbf{y}}) \leq F^\lambda(\mathbf{x}, \mathbf{y}) + \sum_{i=1}^N \lambda_i \langle \tilde{y}_i - y_i, C_i(y_i) - x_i \rangle + \frac{1}{2}(\max_i \frac{\lambda_i}{K_i}) \sum_{i=1}^N (\|\tilde{y}_i - y_i\|_i^*)^2$,
   *where $\|\cdot\|_i^*$ is the dual norm of $\|\cdot\|_i$ (i.e. $\|y_i\|_i^* = \max_{\|x_i\|_i \leq 1} \langle x_i, y_i \rangle$).*

*Remark* 4.1. Collecting each individual choice map into a vector, we obtain the aggregate choice map $C : \mathbb{R}^{\sum_{i=1}^N d_i} \to \mathcal{X}$, with $C(\mathbf{y}) = (C_1(y_1), \dots, C_N(y_N))$. Since each space $\mathcal{X}_i$ is endowed with norm $\|\cdot\|_i$, we can define the induced aggregate norm $\|\cdot\|$ on the joint space $\mathcal{X}$ as follows: $\|\mathbf{x}\| = \sum_{i=1}^N \|x_i\|_i$. We can also similarly define the aggregate dual norm: $\|\mathbf{y}\|^* = \sum_{i=1}^N \|y_i\|_i^*$. Henceforth, it shall be clear that the convergence in the joint space (e.g. $C(\mathbf{y}^t) \to \mathbf{x}, \mathbf{y}^t \to \mathbf{y}$) will be defined under the respective aggregate norm.

Finally, we assume throughout the paper that the choice maps are regular in the following (very weak) sense: a choice map $C(\cdot)$ is said to be $\lambda$-Fenchel coupling conforming if

$$C(\mathbf{y}^t) \to \mathbf{x} \text{ implies } F^\lambda(\mathbf{x}, \mathbf{y}^t) \to 0 \text{ as } t \to \infty. \tag{4.1}$$

Unless one aims for relatively pathological cases, choice maps induced by typical regularizers are always $\lambda$-Fenchel coupling conforming: examples include the Euclidean and entropic regularizers.

## 4.2 Convergence of OMD to Nash Equilibrium

We start by characterizing the assumption on the delay model:

**Assumption 1.** The delays are assumed to be:

1. Synchronous: $\mathcal{G}_i^t = \mathcal{G}_j^t, \forall i, j, \forall t$.

2. Bounded: $d_i^t \leq D, \forall i, \forall t$ (for some positive integer $D$).

**Theorem 4.3.** *Fix a game with continuous action spaces $(\mathcal{N}, \mathcal{X} = \prod_{i=1}^N \mathcal{X}_i, \{u_i\}_{i=1}^N)$ that admits $\mathbf{x}^*$ as the unique Nash equilibrium that is $\lambda$-variationally stable. Under Assumption 1, the OMD iterate $\mathbf{x}^t$ given in Algorithm 1 converges to $\mathbf{x}^*$, irrespective of the initial point $\mathbf{x}^0$.*

*Remark* 4.2. The proof is rather long and involved. To aid the understanding and enhance the intuition, we break it down into four main steps, each of which will be proved in the appendix in detail.

1. Since the delays are synchronous, we denote by $\mathcal{G}^t$ the common set and $d^t$ the common delay at round $t$. The gradient update in OMD under delays can then be written as:

$$y_i^{t+1} = y_i^t + \alpha^t \sum_{s \in \mathcal{G}^t} v_i(\mathbf{x}^s) = y_i^t + \alpha^t \left\{ |\mathcal{G}^t| v_i(\mathbf{x}^t) + \sum_{s \in \mathcal{G}^t} \{v_i(\mathbf{x}^s) - v_i(\mathbf{x}^t)\} \right\}. \tag{4.2}$$

Define $b_i^t = \sum_{s \in \mathcal{G}^t} \{v_i(\mathbf{x}^s) - v_i(\mathbf{x}^t)\}$. We show $\lim_{t \to \infty} \|b_i^t\|_i^* = 0$ for each player $i$.

2. Define $\mathbf{b}^t = (b_1^t, \dots, b_N^t)$ and we have $\lim_{t \to \infty} \mathbf{b}^t = 0$ per Claim 1. Since each player's gradient update can be written as $y_i^{t+1} = y_i^t + \alpha^t (|\mathcal{G}^t| v_i(\mathbf{x}^t) + b_i^t)$ per Claim 1, we can then write the joint OMD update (of all players) as:

$$\mathbf{x}^t = C(\mathbf{y}^t), \tag{4.3}$$
$$\mathbf{y}^{t+1} = \mathbf{y}^t + \alpha^t \{|\mathcal{G}^t| v(\mathbf{x}^t) + \mathbf{b}^t\}. \tag{4.4}$$

Let $B(\mathbf{x}^*, \epsilon) \triangleq \{\mathbf{x} \in \mathcal{X} \mid \|\mathbf{x} - \mathbf{x}^*\| < \epsilon\}$ be the open ball centered around $\mathbf{x}^*$ with radius $\epsilon$. Then, using $\lambda$-Fenchel coupling as a "energy" function and leveraging the handle on $\mathbf{b}^t$ given by Claim 1, we can establish that, for any $\epsilon > 0$ the iterate $\mathbf{x}^t$ will eventually enter $B(\mathbf{x}^*, \epsilon)$ and visit $B(\mathbf{x}^*, \epsilon)$ infinitely often, no matter what the initial point $\mathbf{x}^0$ is. Mathematically, the claim is that $\forall \epsilon > 0, \forall \mathbf{x}^0, |\{t \mid \mathbf{x}^t \in B(\mathbf{x}^*, \epsilon)\}| = \infty$.

3. Fix any $\delta > 0$ and consider the set $\tilde{B}(\mathbf{x}^*, \delta) \triangleq \{C(\mathbf{y}) \mid F^\lambda(\mathbf{x}^*, \mathbf{y}) < \delta\}$. In other words, $\tilde{B}(\mathbf{x}^*, \delta)$ is some "neighborhood" of $\mathbf{x}^*$, which contains every $\mathbf{x}$ that is an image of some $\mathbf{y}$ (under the choice map $C(\cdot)$) that is within $\delta$ distance of $\mathbf{x}^*$ under the $\lambda$-Fenchel coupling "metric". Although $F^\lambda(\mathbf{x}^*, \mathbf{y})$ is not a metric, $\tilde{B}(\mathbf{x}^*, \delta)$ contains an open ball within it. Mathematically, the claim is that for any $\delta > 0$, $\exists \epsilon(\delta) > 0$ such that: $B(\mathbf{x}^*, \epsilon) \subset \tilde{B}(\mathbf{x}^*, \delta)$.

4. For any "neighborhood" $\tilde{B}(\mathbf{x}^*, \delta)$, after long enough rounds, if $\mathbf{x}^t$ ever enters $\tilde{B}(\mathbf{x}^*, \delta)$, it will be trapped inside $\tilde{B}(\mathbf{x}^*, \delta)$ thereafter. Mathematically, the claim is that for any $\delta > 0$, $\exists T(\delta)$, such that for any $t \geq T(\delta)$, if $\mathbf{x}^t \in \tilde{B}(\mathbf{x}^*, \delta)$, then $\mathbf{x}^{\tilde{t}} \in \tilde{B}(\mathbf{x}^*, \delta), \forall \tilde{t} \geq t$.

Putting all four elements above together, we note that the significance of Claim 3 is that, since the iterate $\mathbf{x}^t$ will enter $B(\mathbf{x}^*, \epsilon)$ infinitely often (per Claim 2), $\mathbf{x}^t$ must enter $\tilde{B}(\mathbf{x}^*, \delta)$ infinitely often. It therefore follows that, per Claim 4, starting from iteration $t$, $\mathbf{x}^t$ will remain in $\tilde{B}(\mathbf{x}^*, \delta)$. Since this is true for any $\delta > 0$, we have $F^\lambda(\mathbf{x}^*, \mathbf{y}^t) \to 0$ as $t \to \infty$. Per Statement 1 in Lemma 4.2, this leads to that $\|C(\mathbf{y}^t) - \mathbf{x}^*\| \to 0$ as $t \to \infty$, thereby establishing that $\mathbf{x}^t = C(\mathbf{y}^t) \to \mathbf{x}^*$ as $t \to 0$.

In fact, the result generalizes straightforwardly to multiple Nash equilibria. The proof of the convergence to the set case is line-by-line identical, provided we redefine, in a standard way, every quantity that measures the distance between two points to the corresponding quantity that measures the distance between a point and a set (by taking the infimum over the distances between the point and a point in that set). We directly state the result below.

**Theorem 4.4.** *Fix a game with continuous action spaces $(\mathcal{N}, \mathcal{X} = \prod_{i=1}^N \mathcal{X}_i, \{u_i\}_{i=1}^N)$ that admits $\mathcal{X}^*$ as a $\lambda$-variationally stable set (of necessarily all Nash equilibria), for some $\lambda \in \mathbf{R}_{++}^N$. Under Assumption 1, the OMD iterate $\mathbf{x}^t$ given in Algorithm 1 satisfies $\lim_{t \to \infty} dist(\mathbf{x}^t, \mathcal{X}^*) = 0$, irrespective of $\mathbf{x}^0$, where $dist(\cdot, \cdot)$ is the standard point-to-set distance function induced by the norm $\|\cdot\|$.*

## 5 Delayed Mirror Descent: Asynchronous and Unbounded Delays

The synchronous and bounded delay assumption in Assumption 1 is fairly strong. In this section, by a simple modification of OMD, we propose a new learning algorithm called Delayed Mirror Descent (DMD), that allows the last-iterate convergence-to-Nash result to be generalized to cases with arbitrary asynchronous delays among players as well as unbounded delay growth.

### 5.1 Delayed Mirror Descent in Games

The main idea for the modification is that when player $i$ doesn't receive any gradient on round $t$, instead of not doing any gradient updates as in OMD, he uses the most recent set of gradients to perform updates. More formally, define the most recent information set[6] as:

$$\tilde{\mathcal{G}}_i^t = \begin{cases} \mathcal{G}_i^t, & \textbf{if } \mathcal{G}_i^t \neq \emptyset \\ \tilde{\mathcal{G}}_i^{t-1}, & \textbf{if } \mathcal{G}_i^t = \emptyset. \end{cases}$$

Under this definition, Delayed Mirror Descent is (note that $\tilde{\mathcal{G}}_i^t$ is always non-empty here):

We only make the following assumption on the delays:

**Assumption 2.** For each player $i$, $\lim_{t \to \infty} \sum_{s=\min \tilde{\mathcal{G}}_i^t}^t \alpha^s = 0$.

This assumption essentially requires that no player's delays grow too fast. Note that in particular, players delays can be arbitrarily asynchronous. To make this assumption more concrete, we next give two more explicit delay conditions that satisfy the main delay assumption. As made formal by the following lemma, if the delays are bounded (but not necessarily synchronous), then Assumption 2 is satisfied. Furthermore, by appropriately choosing the sequence $\alpha^t$, Assumption 2 can accommodate delays that are unbounded and grow super-linearly.

**Algorithm 2** Delayed Mirror Descent on Games

---

1: Each player $i$ chooses an initial $y_i^0$.
2: **for** $t = 0, 1, 2, \ldots$ **do**
3:     **for** $i = 1, \ldots, N$ **do**
4:         $x_i^t = \arg\max_{x_i \in \mathcal{X}_i} \{\langle y_i^t, x_i \rangle - h_i(x_i)\}$
5:         $y_i^{t+1} = y_i^t + \frac{\alpha^t}{|\tilde{\mathcal{G}}_i^t|} \sum_{s \in \tilde{\mathcal{G}}_i^t} v_i(\mathbf{x}^s)$
6:     **end for**
7: **end for**

---

**Lemma 5.1.** *Let $\{d_i^s\}_{s=1}^\infty$ be the delay sequences for player $i$.*

1. *If each player $i$'s delay is bounded (i.e. $\exists d \in \mathbb{Z}, d_i^s \leq d, \forall s$), then Assumption 2 is satisfied for any positive, non-increasing, not-summable-but-square-summable sequence $\{\alpha^t\}$.*

2. *There exists a positive, non-increasing, not-summable-but-square-summable sequence (e.g. $\alpha^t = \frac{1}{t \log t \log\log t}$) such that if $d_i^s = O(s \log s), \forall i$, then Assumption 2 is satisfied.*

*Proof:* We will only prove Statement 2, the more interesting case. Take $\alpha^t = \frac{1}{t \log t \log\log t}$, which is obviously positive, non-increasing and square-summable. Since $\int_{s=4}^t \frac{1}{s \log s \log\log s} ds = \log\log\log t \to \infty$ as $t \to \infty$, $\alpha^t$ is not summable. Next, let $\tilde{\mathcal{G}}_i^t$ be given and let $\tilde{t}$ be the most recent round (up to and including $t$) such that $\mathcal{G}_i^{\tilde{t}}$ is not empty. This means:

$$\tilde{\mathcal{G}}_i^t = \mathcal{G}_i^{\tilde{t}}, \mathcal{G}_i^k = \emptyset, \forall k \in (\tilde{t}, t]. \tag{5.1}$$

Note that since the gradient at time $\tilde{t}$ will be available at time $\tilde{t} + d_i^{\tilde{t}} - 1$, it follows that

$$t - \tilde{t} \leq d_i^{\tilde{t}}. \tag{5.2}$$

Note that this implies $\tilde{t} \to \infty$ as $t \to \infty$, because otherwise, $\tilde{t}$ is bounded, leading to the right-side $d_i^{\tilde{t}}$ being bounded, which contradicts to the left-side diverging to infinity.

Since $d_i^s = O(s \log s)$, it follows that $d_i^s \leq K s \log s$ for some $K > 0$. Consequently, Equation 5.2 implies: $t \leq \tilde{t} + K\tilde{t} \log \tilde{t}$. Denote $s_{\min}^t = \min \tilde{\mathcal{G}}_i^t$, Equation 5.1 implies that $s_{\min}^t = \min \mathcal{G}_i^{\tilde{t}}$, thereby yielding $s_{\min}^t + d_i^{s_{\min}^t} - 1 = \tilde{t}$. Therefore:

$$d_i^{s_{\min}^t} = \tilde{t} - s_{\min}^t + 1. \tag{5.3}$$

Equation (5.3) implies that $s_{\min}^t \to \infty$ as $t \to \infty$, because otherwise, the left-hand side of Equation (5.3) is bounded while the right-hand side goes to infinity (since $\tilde{t} \to \infty$ as $t \to \infty$ as established earlier).

With the above notation, it follows that:

$$\lim_{t\to\infty} \sum_{s=\min \tilde{\mathcal{G}}_i^t}^t \alpha^s \leq \lim_{t\to\infty} \sum_{s=s_{\min}^t}^t \alpha^s = \lim_{t\to\infty} \left\{ \sum_{s=s_{\min}^t}^{\tilde{t}} \alpha^s + \sum_{s=\tilde{t}+1}^t \alpha^s \right\} \tag{5.4}$$

$$\leq \lim_{t\to\infty} \left\{ d_i^{s_{\min}^t} \alpha^{s_{\min}^t} + (\tilde{t} \log \tilde{t}) \alpha^{\tilde{t}} \right\} \tag{5.5}$$

$$= \lim_{t\to\infty} \left\{ \frac{d_i^{s_{\min}^t}}{(s_{\min}^t) \log(s_{\min}^t) \log\log(s_{\min}^t)} + \frac{K\tilde{t} \log \tilde{t}}{(\tilde{t}+1) \log(\tilde{t}+1) \log\log(\tilde{t}+1)} \right\} \tag{5.6}$$

$$\leq \lim_{t\to\infty} \left\{ \frac{K(s_{\min}^t) \log(s_{\min}^t)}{(s_{\min}^t) \log(s_{\min}^t) \log\log(s_{\min}^t)} + \frac{K\tilde{t} \log \tilde{t}}{(\tilde{t}+1) \log(\tilde{t}+1) \log\log(\tilde{t}+1)} \right\} \tag{5.7}$$

$$\leq \lim_{t\to\infty} \left\{ \frac{K}{\log\log(s_{\min}^t)} + \frac{K}{\log\log(\tilde{t}+1)} \right\} = 0. \tag{5.8}$$

∎

*Remark* 5.1. The proof to the second claim of Lemma 5.1 indicates that one can also easily obtain slightly larger delay growth rates: $O(t \log t \log \log t)$, $O(t \log t \log \log t \log \log \log t)$ and so on, by choosing the corresponding step size sequences. Further, it is conceivable that one can identify meaningfully larger delay growth rates that still satisfy Assumption 2, particularly under more restrictions on the degree of delay asynchrony among the players. We leave that for future work.

## 5.2 Convergence of DMD to Nash Equilibrium

**Theorem 5.2.** *Fix a game with continuous action spaces* $(\mathcal{N}, \mathcal{X} = \prod_{i=1}^{N} \mathcal{X}_i, \{u_i\}_{i=1}^{N})$ *that admits* $\mathbf{x}^*$ *as the unique Nash equilibrium that is* $\lambda$-*variationally stable. Under Assumption 2, the DMD iterate* $\mathbf{x}^t$ *given in Algorithm 2 converges to* $\mathbf{x}^*$*, irrespective of the initial point* $\mathbf{x}^0$*.*

The proof here uses a similar framework as the one in Remark 4.2, although the details are somewhat different. Building on the notation and arguments given in Remark 4.2, we again outline three main ingredients that together establish the result. Detailed proofs are omitted due to space limitation.

1. The gradient update in DMD can be rewritten as:

$$y_i^{t+1} = y_i^t + \frac{\alpha^t}{|\tilde{\mathcal{G}}_i^t|} \sum_{s \in \tilde{\mathcal{G}}_i^t} v_i(\mathbf{x}^s) = y_i^t + \alpha^t v_i(\mathbf{x}^t) + \alpha^t \sum_{s \in \tilde{\mathcal{G}}_i^t} \frac{v_i(\mathbf{x}^s) - v_i(\mathbf{x}^t)}{|\tilde{\mathcal{G}}_i^t|}.$$

   By defining: $b_i^t = \sum_{s \in \tilde{\mathcal{G}}_i^t} \frac{v_i(\mathbf{x}^s) - v_i(\mathbf{x}^t)}{|\tilde{\mathcal{G}}_i^t|}$, we can write player $i$'s gradient update as:

$$y_i^{t+1} = y_i^t + \alpha^t(v_i(\mathbf{x}^t) + b_i^t).$$

   By bounding $b_i^t$'s magnitude using the delay sequence, Assumption 2 allows us to establish that $b_i^t$ has negligible impact over time. Mathematically, the claim is that $\lim_{t \to \infty} \|b_i^t\|_i^* = 0$.

2. The joint DMD update can be written as:

$$\mathbf{x}^t = C(\mathbf{y}^t), \tag{5.9}$$
$$\mathbf{y}^{t+1} = \mathbf{y}^t + \alpha^t(v(\mathbf{x}^t) + \mathbf{b}^t). \tag{5.10}$$

   Here again using $\lambda$-Fenchel coupling as a "energy" function and leveraging the handle on $\mathbf{b}^t$ given by Claim 1, we show that for any $\epsilon > 0$ the iterate $\mathbf{x}^t$ will eventually enter $B(\mathbf{x}^*, \epsilon)$ and visit $B(\mathbf{x}^*, \epsilon)$ infinitely often, no matter what the initial point $\mathbf{x}^0$ is. Furthermore, per Claim 3 in Remark 4.2, $B(\mathbf{x}^*, \epsilon) \subset \tilde{B}(\mathbf{x}^*, \delta)$. This implies that $\mathbf{x}^t$ must enter $\tilde{B}(\mathbf{x}^*, \delta)$ infinitely often.

3. Again using $\lambda$-Fenchel coupling, we show that under DMD, for any "neighborhood" $\tilde{B}(\mathbf{x}^*, \delta)$, after long enough iterations, if $\mathbf{x}^t$ ever enters $\tilde{B}(\mathbf{x}^*, \delta)$, it will be trapped inside $\tilde{B}(\mathbf{x}^*, \delta)$ thereafter.

Combining the above three elements, it follows that under DMD, starting from iteration $t$, $\mathbf{x}^t$ will remain in $\tilde{B}(\mathbf{x}^*, \delta)$. Since this is true for any $\delta > 0$, we have $F^\lambda(\mathbf{x}^*, \mathbf{y}^t) \to 0$ as $t \to \infty$, thereby establishing that $\mathbf{x}^t = C(\mathbf{y}^t) \to \mathbf{x}^*$ as $t \to 0$.

Here again, the result generalizes straightforwardly to the multiple Nash equilibria case (with identical proofs modulo using point-to-set distance metric). We omit the statement.

## 6 Conclusion

We examined a model of game-theoretic learning based on OMD with asynchronous and delayed information. By focusing on games with $\lambda$- stable equilibria, we showed that the sequence of play induced by OMD converges whenever the feedback delays faced by the players are synchronous and bounded. Subsequently, to tackle fully decentralized, asynchronous environments with unbounded feedback delays (possibly growing sublinearly in the game's horizon), we showed that our convergence result still holds under delayed mirror descent, a variant of vanilla OMD that leverages past information even in rounds where no feedback is received. To further enhance the distributed aspect of the algorithm, in future work we intend to focus on the case where the players' gradient input is not only delayed, but also subject to stochastic imperfections – or, taking this to its logical extreme, when players only have observations of their in-game payoffs, and have no gradient information.

## 7 Acknowledgments

Zhengyuan Zhou is supported by Stanford Graduate Fellowship and he would like to thank Walid Krichene and Alex Bayen for stimulating discussions (and their charismatic research style) that have firmly planted the initial seeds for this work. Panayotis Mertikopoulos gratefully acknowledges financial support from the Huawei Innovation Research Program ULTRON and the ANR JCJC project ORACLESS (grant no. ANR–16–CE33–0004–01). Claire Tomlin is supported in part by the NSF CPS:FORCES grant (CNS-1239166).

## Footnotes

[1] In many formulations, a cost function (as opposed to a reward function) is used, in which case such cost functions need to be convex.

[2] Of course, taking $d^t = 1$ yields the classical no-delay setting.

[3]In the supplement, we give two well-known classes of games that satisfy this equilibrium notion.

[4]Note that boldfaced letters are only used to denote joint actions. In particular, $x_i$ is a vector even though it is not boldfaced.

[5]Note that per the last assumption in the definition of a concave game (Definition 2.1), the gradient $\mathbf{v}(\mathbf{x})$ always exists and is a continuous function on the joint action space $\mathcal{X}$.

[6]There may not be any gradient information in the first few rounds due to delays. Without loss of generality, we can always start at the first round when there is non-empty gradient information, or equivalently, assume that some gradient is available at $t = 0$.

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
