[Supplementary Material]

# Supplementary Material for Countering Feedback Delays in Multi-Agent Learning

**Zhengyuan Zhou**
Stanford University
zyzhou@stanford.edu

**Panayotis Mertikopoulos**
Univ. Grenoble Alpes, CNRS, Inria, LIG
panayotis.mertikopoulos@imag.fr

**Nicholas Bambos**
Stanford University
bambos@stanford.edu

**Peter Glynn**
Stanford University
glynn@stanford.edu

**Claire Tomlin**
UC Berkeley
tomlin@eecs.berkeley.edu

## Abstract

Some of the missing proofs in the paper can be found here. All the results are stated again here (although the numberings are different from the main paper).

## 1 Examples of $\lambda$-Variationally Stable Games

Here we give two important classes of games that satisfy the $\lambda$-variational stability criterion. This is by no means a comprehensive list.

1. **Convex Potential Games** A game $\mathcal{G} = (\mathcal{N}, \mathcal{X} = \prod_{i=1}^N \mathcal{X}_i, \{u_i\}_{i=1}^N)$ is called a potential game [1] if there exists a potential function $V : \mathcal{X} \to \mathbf{R}$ such that $u_i(x_i, \mathbf{x}_{-i}) - u_i(\tilde{x}_i, \mathbf{x}_{-i}) = V(x_i, \mathbf{x}_{-i}) - V_i(\tilde{x}_i, \mathbf{x}_{-i}), \forall i \in \mathcal{N}, \forall \mathbf{x} \in \mathcal{X}, \forall \tilde{x}_i \in \mathcal{X}_i$. A potential game is called a convex potential game if the potential function $V(\cdot)$ is concave[1] Note that in a convex potential game, we have

$$H_{ij}^\lambda(\mathbf{x}) = \frac{1}{2}\lambda_i \nabla_{x_j} v_i(\mathbf{x}) + \frac{1}{2}\lambda_j (\nabla_{x_i} v_j(\mathbf{x}))^T \tag{1.1}$$

$$= \frac{1}{2}\lambda_i \nabla_{x_j} \nabla_{x_i} V(\mathbf{x}) + \frac{1}{2}\lambda_j (\nabla_{x_i} \nabla_{x_j} V(\mathbf{x}))^T. \tag{1.2}$$

   Setting $\lambda = \mathbf{1}$, we obtain $H^{\mathbf{1}}(\mathbf{x}) = \nabla^2 V$, which is negative semi-definite when $V$ is concave. This implies that in a convex potential game, $\mathcal{C} = \arg\max_{\mathbf{x} \in \mathcal{X}} V(\mathbf{x})$ is $\mathbf{1}$-variationally stable per Lemma 2.2.

2. **Linear Cournot Oligopoly Games** There is a set $\mathcal{N} = \{1, 2, \ldots, N\}$ of firms that supply the market with the same good (or service). Firm $i$ provides $x_i \in [0, C_i]$ quantity to the market. The price of the good is a decreasing function of the total quantity of the good supplied to the market: $P(\mathbf{x}) = P(\sum_{i=1}^N x_i)$. A common price function takes the linear form: $P(\sum_{i=1}^N x_i) = a - b(\sum_{i=1}^N x_i), a > 0, b > 0$. The utility function for firm $i$ is then $u_i(\mathbf{x}) = x_i P(\sum_{i=1}^N x_i) - c_i x_i$, where $c_i$ is the unit production cost for firm $i$. In this case, one can clearly that this is a concave game. Further, again set setting $\lambda = \mathbf{1}$, we have

$$H_{ij}^{\mathbf{1}}(\mathbf{x}) = \frac{1}{2}\frac{\partial v_i(\mathbf{x})}{\partial x_j} + \frac{1}{2}\frac{\partial v_j(\mathbf{x})}{\partial x_i} \tag{1.3}$$

$$= -b\delta_{ij} - b, \tag{1.4}$$

where $\delta_{ij} = 1$ if $i = j$ and $\delta_{ij} = 0$ otherwise. Consequently,

$$H_{ij}^{\mathbf{1}}(\mathbf{x}) = -b(\mathbf{I} + \mathbf{1}_{N \times N}),$$

which is negative definite. Hence, the game admits a unique Nash equilibrium that is **1**-variationally stable per Lemma 2.2.

## 2 $\lambda$-Variational Stability: A Key Criterion

Before proceeding, a word on the notation for the remainder of the supplementary material: for convenience, we shall write $v_j(\mathbf{x})(x_j - x_j^*)$ to denote the inner product between $v_j(\mathbf{x})$ and $x_j - x_j^*$ in replacement of the more cumbersome notation $\langle v_j(\mathbf{x}), x_j - x_j^* \rangle$.

**Lemma 2.1.** *If $\mathcal{C}$ is nonempty and $\lambda$-stable, then it is closed, convex and contains all Nash equilibria of the game.*

*Proof.* First we show that any element $\mathbf{x}^* \in \mathcal{C}$ is a Nash equilibrium. For any $i \in \mathcal{N}$, take any $x_i \in \mathcal{X}_i$ and any $\tau \in (0, 1]$, set $\mathbf{x} \triangleq (x_1^*, \dots, x_{i-1}^*, (1-\tau)x_i^* + \tau x_i, x_{i+1}^*, \dots, x_N^*) = \mathbf{x}^* + \tau(x_i - x_i^*)\mathbf{e}_i$, where $\mathbf{e}_i$ is the $j$-th unit vector in the standard basis. By convexity of $\mathcal{X}_i$, $\mathbf{x} \in \mathcal{X}$. Further,

$$\frac{d}{d\tau} u_i(x_i^* + \tau(x_i - x_i^*); \mathbf{x}_{-i}) = v_i(\mathbf{x})(x_i - x_i^*). \tag{2.1}$$

By applying the variational stability condition to the profiles $\mathbf{x}^*$ and $\mathbf{x}$, it follows that the RHS of the above equation is strictly negative for all $\tau > 0$. In turn, this implies that $u_i(\mathbf{x}) \leq u_i(\mathbf{x}^*)$, i.e. $\mathbf{x}^*$ is a Nash equilibrium.

Next, we show that $\mathcal{C}$ is closed. Take any convergent sequence $\{\mathbf{x}^j\}_{j=0}^{\infty}$ in $\mathcal{C}$: $\mathbf{x}^j \in \mathcal{C}, \lim_{j \to \infty} \mathbf{x}^j = \mathbf{x}^*$. Then, for any $\mathbf{x} \in \mathcal{X}$, we have $\sum_{i=1}^{N} \lambda_i v_i(\mathbf{x})(x_i - x_i^j) \leq 0, \forall j = 0, 1, \dots$. Therefore, by continuity, it follows that $\lim_{j \to \infty} \sum_{i=1}^{N} \lambda_i v_i(\mathbf{x})(x_i - x_i^j) = \sum_{i=1}^{N} \lambda_i v_i(\mathbf{x})(x_i - x_i^*) \leq 0, \forall \mathbf{x} \in \mathcal{C}$, thereby implying $\mathbf{x}^* \in \mathcal{C}$. Since $\{\mathbf{x}^j\}_{j=0}^{\infty}$ is any sequence in $\mathcal{C}$, $\mathcal{C}$ contains all its limit points and is therefore closed.

To see that $\mathcal{C}$ is convex, take any $\mathbf{x}^*, \mathbf{y}^* \in \mathcal{C}$ and any $\tau \in [0, 1]$. For any $\mathbf{x} \in \mathcal{X}$, we have

$$\sum_{i=1}^{N} \lambda_i v_i(\mathbf{x})(x_i - (\tau x_i^* - (1-\tau)y_i^*)) = \tag{2.2}$$

$$\tau \sum_{i=1}^{N} \lambda_i v_i(\mathbf{x})(x_i - x_i^*) + (1-\tau) \sum_{i=1}^{N} \lambda_i v_i(\mathbf{x})(x_i - y_i^*) \leq 0, \tag{2.3}$$

thereby establishing that $\tau \mathbf{x}^* + (1-\tau)\mathbf{y}^* \in \mathcal{C}$.

Finally, to see that $\mathcal{C}$ contains all Nash equilibria of the game, assume that $\mathbf{z}^* \notin \mathcal{C}$ is a Nash equilibrium. We then have:

$$\sum_{i=1}^{N} \lambda_i v_i(\mathbf{z}^*)(x_i - z_i^*) \leq 0, \forall \mathbf{x} \in \mathcal{X}. \tag{2.4}$$

Take an arbitrary $\mathbf{x}^* \in \mathcal{C}$. Since $\mathcal{C}$ is $\lambda$-variational stable and $\mathbf{z}^* \notin \mathcal{C}$, we have $\sum_{i=1}^{N} \lambda_i v_i(\mathbf{z}^*)(z_i^* - x_i^*) < 0$, implying that $\sum_{i=1}^{N} \lambda_i v_i(\mathbf{z}^*)(x_i^* - z_i^*) > 0$, which contradicts Equation 2.4. $\square$

**Lemma 2.2.** *Consider a game with continuous actions $(\mathcal{N}, \mathcal{X} = \prod_{i=1}^{N} \mathcal{X}_i, \{u_i\}_{i=1}^{N})$, where each $u_i$ is twice continuously differentiable. For each $\mathbf{x} \in \mathcal{X}$, define the $\lambda$-weighted Hessian matrix $H^{\lambda}(\mathbf{x})$ as follows:*

$$H_{ij}^{\lambda}(\mathbf{x}) = \frac{1}{2}\lambda_i \nabla_{x_j} v_i(\mathbf{x}) + \frac{1}{2}\lambda_j (\nabla_{x_i} v_j(\mathbf{x}))^T. \tag{2.5}$$

*If $H^{\lambda}(\mathbf{x})$ is negative-definite for every $\mathbf{x} \in \mathcal{X}$, then the game admits a unique Nash equilibrium $\mathbf{x}^*$ that is $\lambda$-globally variational stable.*

*Proof.* Per Thereom 6 of [3], it follows that

$$\sum_{i=1}^{N} \lambda_i (v_i(\mathbf{x}) - v_i(\tilde{\mathbf{x}}))(x_i - \tilde{x}_i) \le 0, \forall \mathbf{x}, \tilde{\mathbf{x}} \in \mathcal{X}, \tag{2.6}$$

where equality holds if and only if $\mathbf{x} = \tilde{\mathbf{x}}$. Per Theorem 2 of [3], this inequality then implies that there exists a unique Nash equilibrium $\mathbf{x}^*$. Plug $\mathbf{x}^*$ into Inequality 2.6 for $\tilde{\mathbf{x}}$, we have that for any $\mathbf{x} \in \mathcal{X}$:

$$\sum_{i=1}^{N} \lambda_i v_i(\mathbf{x})(x_i - x_i^*) \le \sum_{i=1}^{N} \lambda_i v_i(\mathbf{x}^*)(x_i - x_i^*) \le 0,$$

where the second inequality follows from $\mathbf{x}^*$ is a Nash equilibrium. Furthermore, both equality are achieved if and only if $\mathbf{x} = \mathbf{x}^*$. This implies that $\{\mathbf{x}^*\}$ is $\lambda$-variational stable. $\square$

## 3 Convergence under Synchronous and Bounded Delays

---
**Algorithm 1** Online Mirror Descent on Games under Delays

---
1: Each player $i$ chooses an initial $y_i^0$.
2: **for** $t = 0, 1, 2, \ldots$ **do**
3:    **for** $i = 1, \ldots, N$ **do**
4:       $x_i^t = \arg\max_{x_i \in \mathcal{X}_i}\{< y_i^t, x_i > -h_i(x_i)\}$
5:       $y_i^{t+1} = y_i^t + \alpha^t \sum_{s \in \mathcal{G}_i^t} v_i(\mathbf{x}^s)$
6:    **end for**
7: **end for**

---

**Assumption 1.** The delays are assumed to be:

1. Synchronous: $\mathcal{G}_i^t = \mathcal{G}_j^t, \forall i, j, \forall t$.

2. Bounded: $d_i^t \le D, \forall i, \forall t$ (for some positive integer $D$).

**Lemma 3.1.** *For each $i \in \{1, \ldots, N\}$, let $h_i : \mathcal{X}_i \to \mathbb{R}$ be a regularizer with respect to the norm $\|\cdot\|_i$ that is $K_i$-strongly convex and let $\lambda \in \mathbb{R}_{++}^N$. Then $\forall \mathbf{x} \in \mathcal{X}, \forall \tilde{\mathbf{y}}, \mathbf{y} \in \mathbb{R}^{\sum_{i=1}^N d_i}$:*

1. *$F^\lambda(\mathbf{x}, \mathbf{y}) \ge \frac{1}{2} \sum_{i=1}^{N} K_i \lambda_i \|C_i(y_i) - x_i\|_i^2 \ge \frac{1}{2}(\min_i K_i \lambda_i) \sum_{i=1}^{N} \|C_i(y_i) - x_i\|_i^2$.*

2. *$F^\lambda(\mathbf{x}, \tilde{\mathbf{y}}) \le F^\lambda(\mathbf{x}, \mathbf{y}) + \sum_{i=1}^{N} \lambda_i \langle \tilde{y}_i - y_i, C_i(y_i) - x_i \rangle + \frac{1}{2}(\max_i \frac{\lambda_i}{K_i}) \sum_{i=1}^{N}(\|\tilde{y}_i - y_i\|_i^*)^2$, where $\|\cdot\|_i^*$ is the dual norm of $\|\cdot\|_i$ (i.e. $\|y_i\|_i^* = \max_{\|x_i\|_i \le 1}\langle x_i, y_i \rangle$).*

**Theorem 3.2.** *Consider a game with continuous actions $(\mathcal{N}, \mathcal{X} = \prod_{i=1}^N \mathcal{X}_i, \{u_i\}_{i=1}^N)$ that admits $\mathbf{x}^*$ as the unique Nash equilibrium that is $\lambda$-variationally stable. Under Assumption 1 for the delays, the OMD iterate $\mathbf{x}^t$ given in Algorithm 1 converges to $\mathbf{x}^*$, irrespective of the initial point $\mathbf{x}^0$.*

*Remark* 3.1. We repeat the four main steps in the following remark and prove each of them in detail in order.

1. Since the delays are synchronous, we denote by $\mathcal{G}^t$ the common set and $d^t$ the common delay at round $t$. The gradient update in OMD under delays can then be written as:

$$y_i^{t+1} = y_i^t + \alpha^t \sum_{s \in \mathcal{G}^t} v_i(\mathbf{x}^s) = y_i^t + \alpha^t \left\{ |\mathcal{G}^t| v_i(\mathbf{x}^t) + \sum_{s \in \mathcal{G}^t} \{v_i(\mathbf{x}^s) - v_i(\mathbf{x}^t)\} \right\}. \tag{3.1}$$

Define $b_i^t = \sum_{s \in \mathcal{G}^t}\{v_i(\mathbf{x}^s) - v_i(\mathbf{x}^t)\}$. We show that under the delay assumption (Assumption 1), $\lim_{t \to \infty} \|b_i^t\|_i^* = 0$ for each player $i$.

2. Define $\mathbf{b}^t = (b_1^t, \ldots, b_N^t)$ and we have $\lim_{t \to \infty} \mathbf{b}^t = 0$ per Claim 1. Since each player's gradient update can be written as $y_i^{t+1} = y_i^t + \alpha^t (|\mathcal{G}^t| v_i(\mathbf{x}^t) + b_i^t)$ per Claim 1, we can then write the joint OMD update (of all players) as:

$$\mathbf{x}^t = C(\mathbf{y}^t), \tag{3.2}$$
$$\mathbf{y}^{t+1} = \mathbf{y}^t + \alpha^t \left\{ |\mathcal{G}^t| v(\mathbf{x}^t) + \mathbf{b}^t \right\}. \tag{3.3}$$

Let $B(\mathbf{x}^*, \epsilon) \triangleq \{\mathbf{x} \in \mathcal{X} \mid \|\mathbf{x} - \mathbf{x}^*\| < \epsilon\}$ be the open ball centered around $\mathbf{x}^*$ with radius $\epsilon$. Then, using $\lambda$-Fenchel coupling as a "energy" function and leveraging the handle on $\mathbf{b}^t$ given by Claim 1, we can establish that, for any $\epsilon > 0$ the iterate $\mathbf{x}^t$ will eventually enter $B(\mathbf{x}^*, \epsilon)$ and visit $B(\mathbf{x}^*, \epsilon)$ infinitely often, no matter what the initial point $\mathbf{x}^0$ is. Mathematically, the claim is that $\forall \epsilon > 0, \forall \mathbf{x}^0, |\{t \mid \mathbf{x}^t \in B(\mathbf{x}^*, \epsilon)\}| = \infty$.

3. Fix any $\delta > 0$ and consider the set $\tilde{B}(\mathbf{x}^*, \delta) \triangleq \{C(\mathbf{y}) \mid F^\lambda(\mathbf{x}^*, \mathbf{y}) < \delta\}$. In other words, $\tilde{B}(\mathbf{x}^*, \delta)$ is some "neighborhood" of $\mathbf{x}^*$, which contains every $\mathbf{x}$ that is an image of some $\mathbf{y}$ (under the choice map $C(\cdot)$) that is within $\delta$ distance of $\mathbf{x}^*$ under the $\lambda$-Fenchel coupling "metric". Although $F^\lambda(\mathbf{x}^*, \mathbf{y})$ is not a metric, $\tilde{B}(\mathbf{x}^*, \delta)$ contains an open ball within it. Mathematically, the claim is that for any $\delta > 0, \exists \epsilon(\delta) > 0$ such that: $B(\mathbf{x}^*, \epsilon) \subset \tilde{B}(\mathbf{x}^*, \delta)$.

4. For any "neighborhood" $\tilde{B}(\mathbf{x}^*, \delta)$, after long enough iterations, if $\mathbf{x}^t$ ever enters $\tilde{B}(\mathbf{x}^*, \delta)$, it will be trapped inside $\tilde{B}(\mathbf{x}^*, \delta)$ thereafter. Mathematically, the claim is that for any $\delta > 0$, there exists a $T(\delta)$, such that for any $t \geq T(\delta)$, if $\mathbf{x}^t \in \tilde{B}(\mathbf{x}^*, \delta)$, then $\mathbf{x}^{\tilde{t}} \in \tilde{B}(\mathbf{x}^*, \delta), \forall \tilde{t} \geq t$.

Putting all four elements above together, we note that the significance of Claim 3 is that, since the iterate $\mathbf{x}^t$ will enter $B(\mathbf{x}^*, \epsilon)$ infinitely often (per Claim 2), $\mathbf{x}^t$ must enter $\tilde{B}(\mathbf{x}^*, \delta)$ infinitely often. It therefore follows that, per Claim 4, starting from iteration $t$, $\mathbf{x}^t$ will remain in $\tilde{B}(\mathbf{x}^*, \delta)$. Since this is true for any $\delta > 0$, we have $F^\lambda(\mathbf{x}^*, \mathbf{y}^t) \to 0$ as $t \to \infty$. Per Statement 1 in Lemma 3.1, this leads to that $\|C(\mathbf{y}^t) - \mathbf{x}^*\| \to 0$ as $t \to \infty$, thereby establishing that $\mathbf{x}^t = C(\mathbf{y}^t) \to \mathbf{x}^*$ as $t \to 0$.

*Proof:*

1. We start by fixing some notation. Let $\mathbf{y}^t = (y_1^t, \ldots, y_N^t), \mathbf{x}^t = (x_1^t, \ldots, x_N^t)$ be the iterates generated in Algorithm 1. Since $\mathcal{X}$ is compact and $v(\cdot)$ is continuous, $V_{\max}^i \triangleq \max_{x_i \in \mathcal{X}_i} \|v_i(\mathbf{x})\|_i^* < \infty, V_{\max} \triangleq \max_{\mathbf{x} \in \mathcal{X}} \|v(\mathbf{x})\|^* = \sum_{i=1}^N V_{\max}^i < \infty$. Since each $h_i(\cdot)$ is $K_i$ strongly convex (with respect to $\|\cdot\|_i$), it follows from a well-known result in convex analysis [2] that the choice map $C(\cdot)$ is $\frac{1}{K}$-Lipschitz continuous, where $K \triangleq \min_i K_i$. Finally, since each $v_i(\cdot)$ is Lipschitz continuous, $v(\cdot)$ is Lipschitz continuous as well and denote the Lipschitz constant as $L$.

Since $d^t \leq D, \forall t$, it follows that $|\mathcal{G}^t| \leq D$ and $\min \mathcal{G}^t \geq t - D + 1$, for otherwise at least one gradient comes from $D + 1$ rounds before. Further, per the OMD update (first equality in Equation 3.1), we have:

$$\|\mathbf{y}^{t+1} - \mathbf{y}^t\|^* = \sum_{i=1}^N \|y_i^{t+1} - y_i^t\|_i^* = \sum_{i=1}^N \|\alpha^t \sum_{s \in \mathcal{G}^t} v_i(\mathbf{x}^s)\|_i^* \tag{3.4}$$

$$\leq \alpha^t \sum_{i=1}^N \sum_{s \in \mathcal{G}^t} \|v_i(\mathbf{x}^s)\|_i^* \leq \alpha^t \sum_{i=1}^N |\mathcal{G}^t| V_{\max}^i \leq \alpha^t D V_{\max} \tag{3.5}$$

By definition, we can expand $b_i^t$ as follows:

$$b_i^t = \sum_{s \in \mathcal{G}^t} \{v_i(\mathbf{x}^s) - v_i(\mathbf{x}^t)\} \le \sum_{s \in \mathcal{G}^t} L \|\mathbf{x}^s - \mathbf{x}^t\| = \sum_{s \in \mathcal{G}^t} L \|C(\mathbf{y}^s) - C(\mathbf{y}^t)\| \le \sum_{s \in \mathcal{G}^t} \frac{L}{K} \|\mathbf{y}^s - \mathbf{y}^t\|^* \tag{3.6}$$

$$\le \frac{L}{K} \sum_{s \in \mathcal{G}^t} \left\{ \|\mathbf{y}^s - \mathbf{y}^{s+1}\|^* + \|\mathbf{y}^{s+1} - \mathbf{y}^{s+2}\|^* + \cdots + \|\mathbf{y}^{t-1} - \mathbf{y}^t\|^* \right\} \tag{3.7}$$

$$\le \frac{L}{K} \sum_{s \in \mathcal{G}^t} \left\{ \alpha^s D V_{\max} + \alpha^{s+1} D V_{\max} + \cdots + \alpha^t D V_{\max} \right\} \tag{3.8}$$

$$= \frac{L D V_{\max}}{K} \sum_{s \in \mathcal{G}^t} \sum_{k=s}^{t} \alpha^k \le \frac{L D V_{\max}}{K} |\mathcal{G}^t| \sum_{s=\min \mathcal{G}^t}^{t} \alpha^s \le \frac{L D^2 V_{\max}}{K} \sum_{s=\min \mathcal{G}^t}^{t} \alpha^s \tag{3.9}$$

$$\le \frac{L D^2 V_{\max}}{K} \sum_{s=t-D+1}^{t} \alpha^s \le \frac{L D^3 V_{\max}}{K} \alpha^{t-D+1} \to 0, \text{ as } t \to \infty, \tag{3.10}$$

where the first inequality in Equation 3.6 follows from the fact that $v(\cdot)$ is $L$-Lipschitz continuous, the second inequality in Equation 3.6 follows from the fact that $C(\cdot)$ is $\frac{1}{K}$-Lipschitz continuous, Equation 3.8 follows from Equations 3.4 and 3.5 and the first inequality in Equation 3.9 follows from that $\alpha^t$'s are non-negative and the second inequality in Equation 3.10 follows from $\alpha^t$ is non-increasing. Lastly, the convergence to 0 in Equation 3.10 follows from the fact that $\alpha^t$ is square-summable.

2. Fix an arbitrary $\epsilon > 0$. Assume for contradiction purposes that $\mathbf{x}^t$ only visits $B(\mathbf{x}^*, \epsilon)$ a finite number of times and hence let $t^0 - 1$ be the last time $\mathbf{x}^t$ is in $B(\mathbf{x}^*, \epsilon)$: $\forall t \ge t^0, \mathbf{x}^t \in \mathcal{X} - B(\mathbf{x}^*, \epsilon)$. Since $\mathcal{X} - B(\mathbf{x}^*, \epsilon)$ is a compact set and $v_i(\mathbf{x})$ is continuous in $\mathbf{x}$ and since by assumption $\sum_{i=1}^{N} \lambda_i v_i(\mathbf{x})(x_i - x_i^*) < 0, \forall \mathbf{x} \in \mathcal{X}, \mathbf{x} \ne \mathbf{x}^*$, it follows that there exists a $c_{\max}(\epsilon) < 0$ such that

$$\sum_{i=1}^{N} \lambda_i v_i(\mathbf{x})(x_i - x_i^*) \le c_{\max}(\epsilon), \forall \mathbf{x} \in \mathcal{X} - B(\mathbf{x}^*, \epsilon). \tag{3.11}$$

Per Claim 1, $\lim_{t \to \infty} \mathbf{b}^t = \mathbf{0}$, therefore, $\|\mathbf{b}^t\|^*$ is bounded and we denote $B_{\max} \triangleq \max_t \|\mathbf{b}^t\|^*$. Next denote $R = \max_{\mathbf{x} \in \mathcal{X}} \|\mathbf{x}\|$, $\lambda_{\max} \triangleq \max_i \lambda_i$ and $\beta^t \triangleq \max_i \frac{(\alpha^t)^2 \lambda_i}{2K_i}$ and note that $\sum_{t=1}^{\infty} \beta^t < \infty$. Using Lemma 3.1, we have $\forall t \ge t^0$:

$$F^\lambda(\mathbf{x}^*, \mathbf{y}^{t+1}) = F^\lambda(\mathbf{x}^*, \mathbf{y}^t + \alpha^t \{|\mathcal{G}^t| v(\mathbf{x}^t) + \mathbf{b}^t\}) \le \tag{3.12}$$

$$F^\lambda(\mathbf{x}^*, \mathbf{y}^t) + \sum_{i=1}^{N} \lambda_i (\alpha^t \{|\mathcal{G}^t| v_i(\mathbf{x}^t) + b_i^t\})(C_i(y_i^t) - x_i^*) + \beta^t (\||\mathcal{G}^t| v(\mathbf{x}^t) + \mathbf{b}^t\|^*)^2 = \tag{3.13}$$

$$F^\lambda(\mathbf{x}^*, \mathbf{y}^t) + \alpha^t \left\{ |\mathcal{G}^t| \sum_{i=1}^{N} \lambda_i v_i(\mathbf{x}^t)(x_i^t - x_i^*) + \sum_{i=1}^{N} \lambda_i b_i^t (x_i^t - x_i^*) \right\} + \beta^t (\||\mathcal{G}^t| v(\mathbf{x}^t) + \mathbf{b}^t\|^*)^2 \tag{3.14}$$

$$\le F^\lambda(\mathbf{x}^*, \mathbf{y}^t) + \alpha^t \left\{ |\mathcal{G}^t| c_{\max}(\epsilon) + \lambda_{\max} \|\mathbf{b}^t\|^* \|\mathbf{x}^t - \mathbf{x}^*\| \right\} + \beta^t \left\{ 2(\||\mathcal{G}^t| v(\mathbf{x}^t)\|^*)^2 + 2(\|\mathbf{b}^t\|^*)^2 \right\} \tag{3.15}$$

$$\le F^\lambda(\mathbf{x}^*, \mathbf{y}^t) + \alpha^t \left\{ |\mathcal{G}^t| c_{\max}(\epsilon) + \lambda_{\max} R \|\mathbf{b}^t\|^* \right\} + 2\beta^t (D^2 V_{\max}^2 + B_{\max}^2) \tag{3.16}$$

$$\le F^\lambda(\mathbf{x}^*, \mathbf{y}^{t^0}) + \left( \sum_{k=t^0}^{t} \alpha^k \right) \left\{ \frac{\sum_{k=t^0}^{t} \alpha^k |\mathcal{G}^k|}{\sum_{k=t^0}^{t} \alpha^k} c_{\max}(\epsilon) + \lambda_{\max} R \frac{\sum_{k=t^0}^{t} \alpha^k \|\mathbf{b}^k\|^*}{\sum_{k=t^0}^{t} \alpha^k} \right\} \tag{3.17}$$

$$+ 2 \left( \sum_{k=t^0}^{t} \beta^k \right) (D^2 V_{\max}^2 + B_{\max}^2), \tag{3.18}$$

where Equation (3.15) follows from Equation (3.11) and Equation (3.17) follows from telescoping.

Next, we show that:

$$1 \leq \lim_{t \to \infty} \frac{\sum_{k=t^0}^{t} \alpha^k |\mathcal{G}^k|}{\sum_{k=t^0}^{t} \alpha^k} \leq D. \tag{3.19}$$

Partition the rounds into intervals $\{0, 1, \ldots, D-1\}, \{D, D+1, \ldots, 2D-1\}, \ldots$. Since each gradient is received exactly once with at most delay $D$, the gradients corresponding to the first interval will have been completely received by time $2D - 1$ (i.e. by the end of the second interval). Similarly, the gradients corresponding to the $l$-th interval will have been all received by time $(l+1)D - 1$. Consequently, since $\alpha^t$ is non-increasing, it follows that:

$$\sum_{k=0}^{\infty} \alpha^k |\mathcal{G}^k| \geq \sum_{l=2}^{\infty} D\alpha^{lD-1} \geq \sum_{k=2D-1}^{\infty} \alpha^k = \infty.$$

Consequently,

$$\lim_{t \to \infty} \frac{\sum_{k=t^0}^{t} \alpha^k |\mathcal{G}^k|}{\sum_{k=t^0}^{t} \alpha^k} = \lim_{t \to \infty} \frac{\sum_{k=0}^{t} \alpha^k |\mathcal{G}^k|}{\sum_{k=0}^{t} \alpha^k} = \lim_{t \to \infty} \frac{\sum_{k=0}^{t} \alpha^k |\mathcal{G}^k|}{\sum_{k=2D-1}^{t} \alpha^k} \geq 1.$$

Finally, $\frac{\sum_{k=t^0}^{t} \alpha^k |\mathcal{G}^t|}{\sum_{k=t^0}^{t} \alpha^k} \leq D$ follows easily by noting that $|\mathcal{G}^t| \leq D$.

Next, note that since $\lim_{t \to \infty} \mathbf{b}^t = \mathbf{0}$ and $\sum_{t=0}^{\infty} \alpha^t = \infty$, it follows that:

$$\frac{\sum_{k=t^0}^{t} \alpha^k \|\mathbf{b}^k\|^*}{\sum_{k=t^0}^{t} \alpha^k} \to 0, \text{ as } t \to \infty. \tag{3.20}$$

Combining Equation 3.19 and Equation 3.20, we obtain:

$$\left( \sum_{k=t^0}^{t} \alpha^k \right) \left\{ \frac{\sum_{k=t^0}^{t} \alpha^k |\mathcal{G}^k|}{\sum_{k=t^0}^{t} \alpha^k} c_{\max}(\epsilon) + \lambda_{\max} R \frac{\sum_{k=t^0}^{t} \alpha^k \|\mathbf{b}^k\|^*}{\sum_{k=t^0}^{t} \alpha^k} \right\} \to -\infty, \text{ as } t \to \infty.$$

Since $\sum_{k=t^0}^{\infty} \beta^k < \infty$, Equation (3.17) implies that $F^\lambda(\mathbf{x}^*, \mathbf{y}^t) \to -\infty$, which contradicts the first statement in Lemma 3.1. The claim therefore follows.

3. Assume for contradiction purposes no $B(\mathbf{x}^*, \epsilon)$ is contained in $\tilde{B}(\mathbf{x}^*, \delta)$, which means that for any $\delta > 0, \exists \mathbf{y}^l$, such that $\|Q(\mathbf{y}^l) - \mathbf{x}^*\| = \delta$ but $F^\lambda(\mathbf{x}^*, \mathbf{y}^l) \geq \epsilon$. This produces a sequence $\{\mathbf{y}^l\}_{l=0}^{\infty}$ such that $C(\mathbf{y}^l) \to \mathbf{x}^*$ but $F^\lambda(\mathbf{x}^*, \mathbf{y}^l) \geq \epsilon, \forall l$. This contradicts with the fact that the choice map $C(\cdot)$ is $\lambda$-Fenchel coupling conforming, because by definition it holds that if $C(\mathbf{y}^t) \to x$, then $F^\lambda(x, \mathbf{y}^t) \to 0$. Consequently, the claim follows.

4. Fix a given $\delta > 0$. Since $\beta^t$ is summable and $\alpha^t$ is not summable but square summable, it follows that $\beta^t \to 0, \alpha^t \to 0, \frac{\alpha^t}{\beta^t} \to \infty$ as $t \to \infty$. There, denote

   (a) $T^1(\delta) = \arg \min_t \{t \mid \beta^s < \frac{\delta}{8(D^2 V_{\max}^2 + B_{\max}^2)}, \forall s \geq t\}$.

   (b) $T^2(\delta) = \arg \min_t \{t \mid c_{\max}(\epsilon(\frac{\delta}{2})) < -2\lambda_{\max} R \|\mathbf{b}^s\|^*, \forall s \geq t\}$.

   (c) $T^3(\delta) = \arg \min_t \{t \mid \alpha^s < \frac{\delta}{4\lambda_{\max} R B_{\max}}, \forall s \geq t\}$.

   (d) $T^4(\delta) = \arg \min_t \{t \mid \frac{\alpha^s}{\beta^s} > \frac{4(D^2 V_{\max}^2 + B_{\max}^2)}{-c_{\max}(\epsilon(\frac{\delta}{2}))}, \forall s \geq t\}$.

   We have $T^1(\delta) < \infty, T^2(\delta) < \infty$ (since $\lim_{t \to \infty} \|\mathbf{b}^t\|^* = 0$ and note that $c_{\max}(\epsilon(\frac{\delta}{2})) < 0$ by definition), $T^3(\delta) < \infty, T^4(\delta) < \infty$. Take

$$T(\delta) = \max(T^1(\delta), T^2(\delta), T^3(\delta), T^4(\delta)\}.$$

   Now take any $t \geq T(\delta)$. We show that if $\mathbf{x}^t \in \tilde{B}(\mathbf{x}^*, \delta)$, then $\mathbf{x}^{t+1} \in \tilde{B}(\mathbf{x}^*, \delta)$. To see that this statement holds, let $\mathbf{x}^t \in \tilde{B}(\mathbf{x}^*, \delta)$, which implies that $F^\lambda(\mathbf{x}^*, \mathbf{y}^t) < \delta$. Note that it suffices to consider $\mathcal{G}^t \neq \emptyset$, for otherwise $\mathbf{x}^{t+1} = \mathbf{x}^t$.

   Now there are two possibilities:

(a) Possibility 1: $\mathbf{x}^t \in B(\mathbf{x}^*, \epsilon(\frac{\delta}{2}))$.

(b) Possibility 2: $\mathbf{x}^t \in \tilde{B}(\mathbf{x}^*, \delta) - B(\mathbf{x}^*, \epsilon(\frac{\delta}{2}))$.

Under Possibility 1, it follows

$$F^\lambda(\mathbf{x}^*, \mathbf{y}^{t+1}) \leq F^\lambda(\mathbf{x}^*, \mathbf{y}^t) + \alpha^t \sum_{i=1}^{N} \lambda_i \{|\mathcal{G}^t| v_i(\mathbf{x}^t) + b_i^t\}(x_i^t - x_i^*) + \beta^t (\||\mathcal{G}^t| v(\mathbf{x}_t) + \mathbf{b}^t\|^*)^2$$

(3.21)

$$\leq F^\lambda(\mathbf{x}^*, \mathbf{y}^t) + \alpha^t \sum_{i=1}^{N} \lambda_i b_i^t (x_i^t - x_i^*) + \beta^t \left\{ 2(\||\mathcal{G}^t| v(\mathbf{x}_t)\|^*)^2 + 2(\|\mathbf{b}^t\|^*)^2 \right\} \qquad (3.22)$$

$$\leq F^\lambda(\mathbf{x}^*, \mathbf{y}^t) + \alpha^t \lambda_{\max} R B_{\max} + 2\beta^t (D^2 V_{\max}^2 + B_{\max}^2) \qquad (3.23)$$

$$< F^\lambda(\mathbf{x}^*, \mathbf{y}^t) + \frac{\delta}{4\lambda_{\max} R B_{\max}} \lambda_{\max} R B_{\max} + \frac{2\delta}{8(D^2 V_{\max}^2 + B_{\max}^2)} (D^2 V_{\max^2} + B_{\max}^2)$$

(3.24)

$$\leq \frac{\delta}{2} + \frac{\delta}{4} + \frac{\delta}{4} = \delta, \qquad (3.25)$$

where the second inequality follows from $\lambda$-variational stability and the last inequality follows from the fact that $\mathbf{x}^t \in B(\mathbf{x}^*, \epsilon(\frac{\delta}{2})) \subset \tilde{B}(\mathbf{x}^*, \frac{\delta}{2})$ per Claim 2.

Under Possibility 2, it follows from Equation 3.16 that

$$F^\lambda(\mathbf{x}^*, \mathbf{y}^{t+1}) \leq F^\lambda(\mathbf{x}^*, \mathbf{y}^t) + \alpha^t \left\{ |\mathcal{G}^t| c_{\max}(\epsilon(\frac{\delta}{2})) + \lambda_{\max} R\|\mathbf{b}^t\|^* \right\} + 2\beta^t (D^2 V_{\max}^2 + B_{\max}^2)$$

(3.26)

$$\leq F^\lambda(\mathbf{x}^*, \mathbf{y}^t) + 2\beta^t (D^2 V_{\max}^2 + B_{\max}^2) \left\{ \frac{\alpha^t}{\beta^t} \frac{c_{\max}(\epsilon(\frac{\delta}{2})) + \lambda_{\max} R\|\mathbf{b}^t\|^*}{2(D^2 V_{\max}^2 + B_{\max}^2)} + 1 \right\} \quad (3.27)$$

$$\leq F^\lambda(\mathbf{x}^*, \mathbf{y}^t) + 2\beta^t (V_{\max}^2 + B_{\max}^2) \left\{ \frac{\alpha^t}{\beta^t} \frac{c_{\max}(\epsilon(\frac{\delta}{2}))}{4(V_{\max}^2 + B_{\max}^2)} + 1 \right\} \qquad (3.28)$$

$$< F^\lambda(\mathbf{x}^*, \mathbf{y}^t) < \epsilon, \qquad (3.29)$$

where the second inequality follows from $|\mathcal{G}^t| \geq 1$ since it is not empty by assumption and $c_{\max} < 0$, the third inequality follows from $\lambda_{\max} R\|\mathbf{b}^t\|^* < -\frac{1}{2} c_{\max}(\epsilon(\frac{\delta}{2}))$ since $t \geq T^2(\delta)$ and the second-to-last inequality follows from $\frac{\alpha^t}{\beta^t} \frac{c_{\max}(\epsilon(\frac{\delta}{2}))}{4(V_{\max}^2 + B_{\max}^2)} + 1 < 0$ since $t \geq T^4(\delta)$.

■

## Footnotes

[1]It is called convex potential game as opposed to concave potential game because in engineering, the utility is typically framed in terms of costs and convex costs correspond to concave utilities.