[Reviews · NeurIPS 2017]

Reviewer 1



If we accept that distributed learning is interesting, then this article presents a nice treatment of distributed mirror descent in which feedback may be asynchronous and delayed. Indeed, we are presented with a provably convergent learning algorithm for continuous action sets (in classes of games) even when individual players' feedback are received with differing levels of delay; further more the regret at time T is controlled as a function of the total delay to time T. This is a strong result, achieved by using a suite of very current proof techniques - lambda-Fenchel couplings serving as primula-dual Bregman divergences and associated tools. I have some concerns, but overall I think this is a good paper. - (very minor) In the first para of Section 2.2, "following learning scheme" actually refers to Algorithm 1, over the page. - Lemma 3.2. If the concept of variational stability implies that all Nash equilibria of a game are in a closed and convex set, to me this is a major restriction on the class of games for which the result is relevant. Two example classes are given in the supplementary material, but I think the main paper should be more upfront about what kinds of games we expect the results to hold in. - At the bottom of page 5, we are presented with an assumption (buried in the middle of a paragraph) which is in some sense the converse to Lemma 4.2. While the paper claims the assumption is very weak, I would still prefer it to be made more explicitly, and some more efforts made to explain why it's weak, and what might make it fail - In Algorithm 2, the division by |G^t_i| is very natural. Why is something similar not done in Algorithm 1? - (Minor) Why do we talk about "Last iterate convergence"? This is a term I'm unfamiliar with. I'm more used to "convergence of intended play" or "convergence of actual play". - References 15 and 16 are, I think, repeats? - You should probably be referencing recent works e.g. [Mertikopoulos] and [Bervoets, S., Bravo, M., and Faure, M]

Reviewer 2



The authors show that in continuous games that satisfy a global variational stability property, Online Mirror Descent converges pointwise to the equilibrium set, even under delayed feedback (as long as it is synchronous across players). They also give a variant of OMD that converges even under asynchronous feedback. I think the authors present interesting results, in particular their delayed feedback results are quite interesting and their Fenchel based Lyapunov potential is an interesting technique. For these reasons it deserves acceptance. On the negative side I do feel that games that satisfy global variational stability are fairly limited and as expected, the authors fail to portray many example application (only convex potential games and linear Cournot games, which are classes that are sort of "easy" in terms of convergence of decentralized dynamics). So this limits the scope of these results.

Reviewer 3



The paper considers a repeated multi-player game with convex compact action sets. Each player is assumed to play an Online Mirror Descent policy. The main question is the convergence of this global discrete-time dynamic to an equilibrium. A new notion of stability (for equilibria), called lambda-variational stability, is introduced. It is proved that if the set of Nash equilibria of the game is variationaly stable, then the dynamic converges to the latter set. An additional sophistication is considered: the feedback of each player may be delayed. The main issue that bothers me is the motivation: on p2, "it is natural to assume that players adopt some variant of online mirror descent when faced with such online decision process". Why not some other regret minimizing policy or some random policy ? The only motivation I can think of, for the study of the convergence of such discrete-time game dynamic to equilibra, is the computation of an equilibirum (similarly to optimization problems where the goal is the computation of some minimizer). But then, the derivation of some explicit rate of convergence would be desirable, as well as a discussion about the computational efficiency of the procedure. Besides, the main point of the paper seems to be the notion of lambda-variational stability and the convergence analysis. The introduction of delayed feedbacks feels secondary and a bit distracting. Another issue is the great similarity of the notion of variational stability and the tools used in the convergence analysis (like Fenchel coupling) with e.g. the paper P. Mertikopoulos, Learning in concave games with impecfect information, arxiv, 2016. A precise comparison with (and of course a reference to) the previous paper should have been included in order for the reader to measure the real contribution of the present paper.